# Snowpack as Indicators of Atmospheric Pollution: The Valday Upland

**Marina Dinu \*, Tatyana Moiseenko \***  **and Dmitry Baranov \***

Vernadsky Institute of Geochemistry and Analytical Chemistry, 119334 Moscow, Russia

\* Correspondence: marinadinu999@gmail.com (M.D.); moiseenko.ti@gmail.com (T.M.);
mitya.baranov.90@bk.ru (D.B.)

**Abstract:** Snowpack is a unique indicator in assessing both local and transboundary contaminants. We considered the features of the snow chemical composition of the Valday Upland, Russia, as a location without a direct influence of smelters (conditional background) in 2016–2019. We identified the influence of a number of geochemical (landscape), biological (trees of the forest zone, vegetation), and anthropogenic factors (technogenic elements—lead, nickel) on the formation of snow composition. We found increases in the content of metals of technogenic origin in city snowfall in the snowpack: cadmium, lead, and nickel in comparison with snowfall in the forest. Methods of sequential and parallel membrane filtration (in situ) were used along with ion-exchange separation to determine metal speciation (labile, unlabile, inorganic speciation with low molecular weight, connection with organic ligands) and explain their migration ability. We found that forest snow samples contain metal compounds (Cu, Pb, and Ni) with different molecular weights due to the different contributions of organic substances. According to the results of filtration, the predominant speciation of metals in the urban snow samples is suspension emission (especially more 8 mkm). The buffer abilities of snowfall in the forest (in various landscapes) and in the city of Valday were assessed. Based on statistical analysis, a significant difference in the chemical composition of snow in the forest and in the city, as well as taking into account the landscape, was shown. Snow on an open landscape on a hill is most susceptible to airborne pollution (sulfates, copper, nickel), city snow is most affected by local pollutants (turbidity, lead).

**Keywords:** snowfall; background area; heavy metals; local input and transboundary migration

## 1. Introduction

The contents of various pollutants in the snowpack (copper, nickel, lead, sulfates, nutrients, etc.) are objective indicators of air quality in winter [1]. A number of properties of the snowpack (high ability to absorb, store pollution) make it convenient for understanding the migration of airborne pollutants [2]. If the snow was not subjected to intensive melting, all atmosphere pollutants (heavy metals, nitrogen oxides, sulfates) would be actively accumulated and saved in various concentrations depending on the weather conditions. Snowpack contains two to three orders more pollutants than other atmospheric precipitation [1–3].

Technogenic components—metal (Cu, Ni, and Cd) ions and anions ($SO_4^{2-}$ and $NO_3^-$)—are of significant interest for atmospheric pollution assessment. Pollutants are characterized by different sources—located within the same place and as transboundary transfers from other locations [3].

The research on snow in background areas has allowed us to assess the migration of the most active components. The snowpack in Greenland and the Arctic, which reflect the pollutants' transboundary flows, is an area of background research. Based on chemical and isotopic analysis, a number of authors [4–6] showed that $SO_4^{2-}$ in winter and early spring snowpack may come from both

anthropogenic and natural sources, respectively, covering long distances due to migration. A number of articles also considered the trends in snow chemical composition changes at certain points at a distance from industrial plants [5–7]. These articles indicated the highest possible metal contents, mainly associated with technogenic impacts. The authors attributed especially high contents of Pb and Cu to the effect of urban infrastructure. However, research on unpolluted areas is especially important in protected areas [8].

Along with the distribution of heavy metals, the acidic neutralizing capacity (ANC) indicator is used to determine the buffer capacity of snowfall (i.e., system stability) with respect to pollutants. The balance between cations and anions allows the assessment of acidic or basic domination in the chemical composition and the determination of the main sources of the chemical migration [3,4,8].

Depending on the locations of the sampling sites, many authors have considered both a strong technogenic impact close to the pollution source and at a considerable distance from the source of pollutions [4,5,9]. Background conditions reflecting global and regional flows of pollutants are of interest. The example of small lakes not subjected to direct anthropogenic impacts has proven that air pollution leads to acidification (in acid-vulnerable regions) and increases in heavy metals in water [10].

To assess the impact of air pollution, a conditional reference area, the Valday National Park in Russia, was chosen. Valday National Park is one of the largest specially protected natural sites of European Russia and was formed to preserve the lake–forest complex of the Valday Upland. The territory of the Valday district is located in the taiga zone, composed of a powerful complex of sedimentary rocks that ubiquitously cover the crystalline basement. It is located far from direct sources of pollution; therefore, the chemical composition of the snowpack reflects the natural and region-wide pattern of elements distribution over and precipitation on the underlying surface.

The objectives of the research were to (1) identify the temperature fluctuations in the studied winter period for air and snow cover; (2) study the physicochemical parameters of snow samples in situ and melted snow samples, and the features of the element speciation migration with respect to the landscape; and (3) assess the snowpack pollution as an indicator of the impact of air pollutions in background areas.

## 2. Materials and Methods

### 2.1. Sampling Sites

The selected area is the lack of local sources of pollution (it is a protected area of the Russian Federation). The city of Valday, located more than 100 km away from large industrial centers, does not have any developed industry [8–11]. Local urban impacts and transboundary transfers may influence the formation of the snow's chemical composition [5].

The studies were conducted from 2016–2018. Two sites were selected for sampling (Figure 1). The first is located 20 km away from the city of Valday near the small Lake Gusinoye; the second is close to the city of Valday.

In the first studied area, the landscape is characterized by a slightly sloping terrain with weak podzols (Albic Retisols); the vegetation is represented by coniferous and broad-leaved trees (spruce, birch, pine, alder etc.). We selected 4 sampling points corresponding to various types of elementary landscapes [11]. Sampling point No. 1 was located on a characteristic eluvial landscape with sufficient atmospheric feed and proximity to coniferous trees; no sloping terrain was observed. Sampling point No. 2 was located on the transeluvial landscape with a surface slope angle of 10°–15°; the territory is characterized by sparse vegetation and the proximity of coniferous trees. Sampling point No. 3 was located on a superaqual landscape with a low terrain. This superaqual landscape represents a geomorphological structure (marshes). In the summer, this type of landscape is heavily flooded, water-sick in some places, and vegetation is represented by mosses that are common in water-sick areas. This point is closest to Lake Gusinoye. Point No. 4 represents the subaquatic landscape. The subaquatic landscape is a water body, Lake Gusinoye (210,000 m$^2$), predominantly of atmospheric feeding (the snow

sample was taken from the lake's frozen surface). The terrain of the studied area near Lake Gusinoye formed as a result of the interaction between various processes of prolonged continental denudation and glacial deposition, being glacial-deposit terrain with a predominance of hilly outwash valleys.

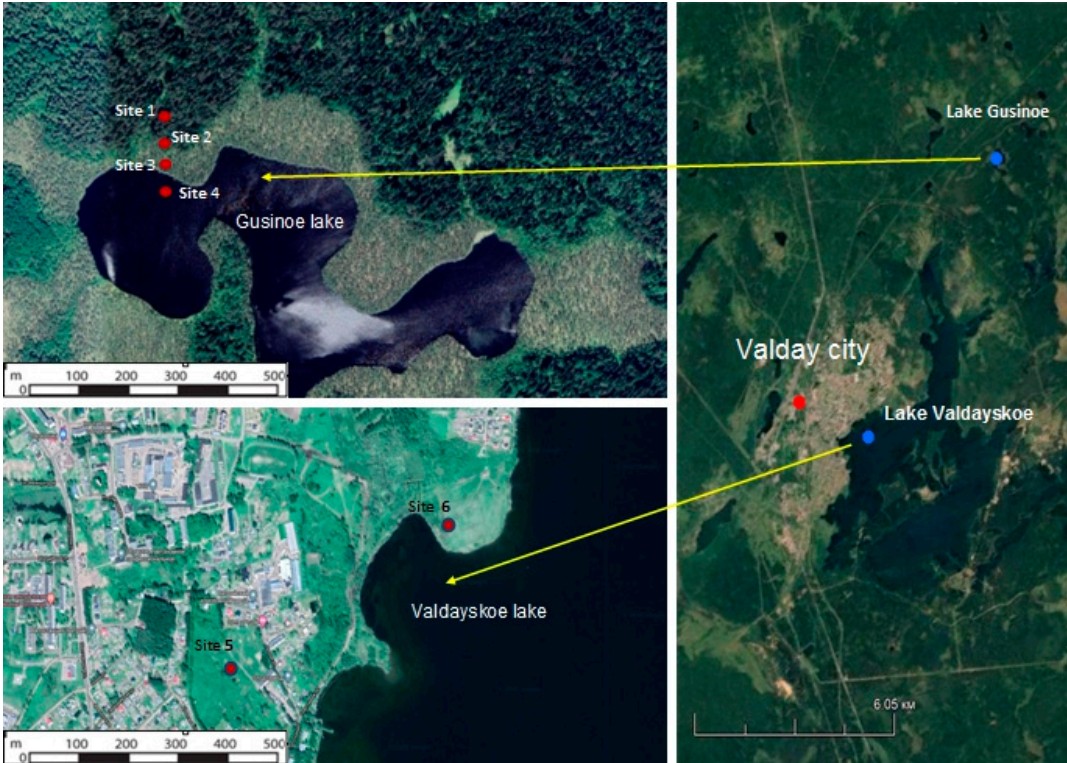

**Figure 1.** Sampling points and filtering equipment.

Samples No. 5 and No. 6 of the second site were located within the territory of the city of Valday at different distances from Lake Valday. Sampling point No. 5 was located on a slightly sloping hill (upland) with sparse vegetation; the soils are highly differentiated compared to soils of the forest territory, which indicates different rock-forming minerals. The landscape is predominantly eluvial. Sampling point No. 6 represents a flat lake plain close to Lake Valday, with sparse vegetation and a superaquatic landscape.

*2.2. Sample Collection and Preparation*

2.2.1. Chemical Analysis of Samples

Snow samples were taken (every March) with a special 1 m high cylindrical sampler; the sampling surface diameter was 0.25 m. To obtain a representative sample, the envelope method was used to collect the combined sample. When sampling, the height of the snow column and a number of physicochemical parameters of the initial snow were measured. In snow samples, pH and temperature (T) were measured (in situ) using a pH Mettler Tolledo instrument (Mettler Tolledo Switzerland) with electrodes for emulsions.

Within 10–15 min, the samples were transported to the laboratory in chemically resistant plastic containers and melted naturally for 24–30 h in covered containers.

The micro-element content in the melted snow samples was analyzed in filtered (0.45 μm according to hydrochemical standards) and unfiltered solutions by inductively coupled plasma atomic emission spectrometry (iCAP-6500, Thermo Scientific, Waltham, MA, USA) and inductively coupled plasma mass spectrometry (X-7, Thermo Scientific). The reliability of the results was confirmed by the convergence of the material balance and electrical neutrality of the system.

Hydrochemical parameters were analyzed within 48 h after the preparation of meltwater and included:

(1)　pH, electrical conductivity: potentiometric determination;

(2)　Permanganate index: titrimetric determination;

(3)　Alkalinity: potentiometric titration using the Gran method;

(4)　Color: spectrophotometric determination;

(5)　Nitrates and nitrites: spectrophotometric determination with Griess reagent;

(6)　Ammonium: spectrophotometric determination with Nessler reagent;

(7)　Phosphates: spectrophotometric determination of the P–Mo complex in an acidic medium;

(8)　Anionic–cationic macro-composition (calcium, magnesium, potassium, sodium, ammonium ions; sulfates; chlorides; fluorides; nitrates; nitrites; and ortho-/polyphosphates) using ion chromatography;

(9)　Copper, lead, cadmium, nickel, cobalt, and iron ions using stripping voltammetry; and alkaline-earth metals, aluminum, copper, nickel, cadmium, and cobalt using the atomic absorption method.

To determine the anionic and cationic composition (in-house data verification), we applied the chromatographic method (liquid ion chromatography, Institute of Geochemistry and Analytical Chemistry (GEOKHI RAS) of the Russian Academy for Sciences) using carbonate buffer and sulfuric acid, respectively.

After appropriate sample preparation, a wide range of elements in micro- and nano-concentrations was determined by inductively coupled plasma atomic emission spectrometry (iCAP-6500, Thermo Scientific) and inductively coupled plasma mass spectrometry (X-7, Thermo Scientific) at the Analytical Center of the Federal National State-Funded Scientific Institution of the Institute for Problems of Technology of Microelectronics and High-Purity Materials of RAS, and at IGAC of RAS.

The results of parallel determinations of element concentrations by different methods were compared to verify the reliability of the measurements of specific parameters. Quality control of the obtained data array was based on the electrical neutrality of water samples. The total number of negatively and positively charged particles, expressed in milli- or micro-equivalents per liter, was equal (up to 10% error).

2.2.2. Membrane Filtration and Ion Exchange Separation of Aqueous Samples

Samples fractioning (Figure 2) was performed directly on-site and after their transportation to the laboratory (in a special cooler bag). For the comparison and specification of results, syringe filter nozzles and a multistep tangential membrane filtration unit [11] were used. The membrane pore sizes were: 8 μm, 1.2 μm, 0.45 μm, and 0.2 μm (VLADiSART). The filter holder was SWINNEX Millipor (25 and 47 mm). The sample volume was 250 mL. The initial samples were consequently filtered through a membrane filter holder (SWINNEX Millipor 47 mm) with an 8 μm membrane (VLADiSART) installed. Further, the filtrate was filtered through 1.2 μm (VLADiSART), 0.45 μm (VLADiSART), and 0.2 μm (VLADiSART). Sample volume −250 mL 200–250 mL of the initial sample is successively passed through a membrane filter holder (SWINNEX Millipor 47 mm) with an installed 8 μm membrane (VLADiSART). The filtrate is then passed through a 1.2 μm membrane (VLADiSART), then 0.45 μm (VLADiSART), and 0.2 μm (VLADiSART). A volume of 5 mL was discarded from each filtrate fraction for elemental analysis by inductively coupled plasma mass spectrometry. At the multistep tangential membrane filtration unit, the sample was passed through the same set of membranes and picked the solutions to be analyzed. The mass balance of all metals was monitored at each stage of fractioning. The average metals mass distribution didn't exceed 15%. Notably, the results of metals fractioning with the syringe filter nozzles and the multistep tangential membrane system coincided. The syringe filter nozzles system allows fractioning directly on-site. Nevertheless, its usage becomes difficult if water turbidity increases as compared with the water used during this study. Sample volume −250 mL 200–250 mL of the initial sample is successively passed through a membrane filter holder (SWINNEX Millipor

47 mm) with an installed 8 μm membrane (VLADiSART). The filtrate is then passed through a 1.2 μm membrane (VLADiSART), then 0.45 μm (VLADiSART), and 0.2 μm (VLADiSART). A volume of 5 mL was discarded from each filtrate fraction for elemental analysis. The following filtrate characteristics were used: microfiltration-based, mechanical suspension, and oxidized contaminants (>8 μm, 1.2 μm, 0.45 μm, 0.2 μm).

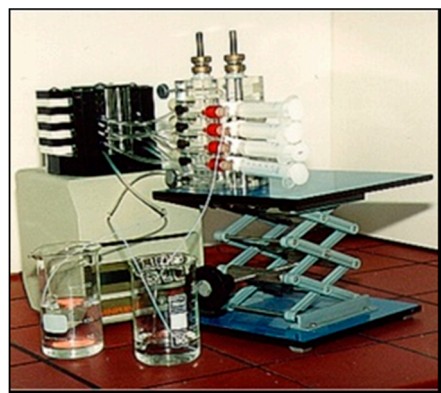

**Figure 2.** Filtration apparatus.

In the course of the experimental work, the metal contents in suspended (unfiltered) and dissolved fractions (filtered) were determined. They were subsequently separated into fixed and non-fixed ones with ion-exchange resin, strongly bound to the organic matter of natural waters (Dowex 50W-X8, 50–100 mesh in Na + form). That is, conditionally labile and unlabile according to the selected ion exchange [12,13]. The labile speciation of elements was aqua-ionic, connected with inorganic (including mixed) metal complexes, and also weakly complex with organic matter (with low conditional stability constant). Unlabile speciation included strong metal complexes with organic matter, mainly of humic nature.

2.2.3. Statistical Analysis

The obtained data were statistically processed with Statistic Advanced 12 software. The discriminant analysis method with a canonical graphic image was used to assess significant differences in the set of parameters. To identify strong relationships between dependent and independent variables, principal component analysis and redundancy analysis principal component analysis (PCA) with redundancy diagram (RDA-x) visualization was applied.

**3. Results and Discussion**

*3.1. Key Seasonal and Physicochemical Parameters*

3.1.1. Influence of Temperature Fluctuations on the Snowpack

From 2016–2019, the average temperature for the period of snowfall (from mid-October to the end of March) ranged from 0.3 to more than 2 °C: −5.3 °C in 2015–2016, −3.4 °C in 2016–2017, −3.9 °C in 2017–2018, and −3.6 °C in 2018–2019. The dynamics of the parameter in each studied year is represented in Figure 3. Notably, the balance between positive and negative temperatures in the selected period varied significantly depending on the year. The highest temperature fluctuations were typical for 2015–2016 and 2016–2017. In the winter of 2016, both maximum positive and minimum negative temperatures were detected. In addition, over the four-year study period, the actual measured temperature (in situ) of snow also increased around 1 °C for each study point. The most obvious changes were observed for forest sampling points of −0.7 to 1.17 °C.

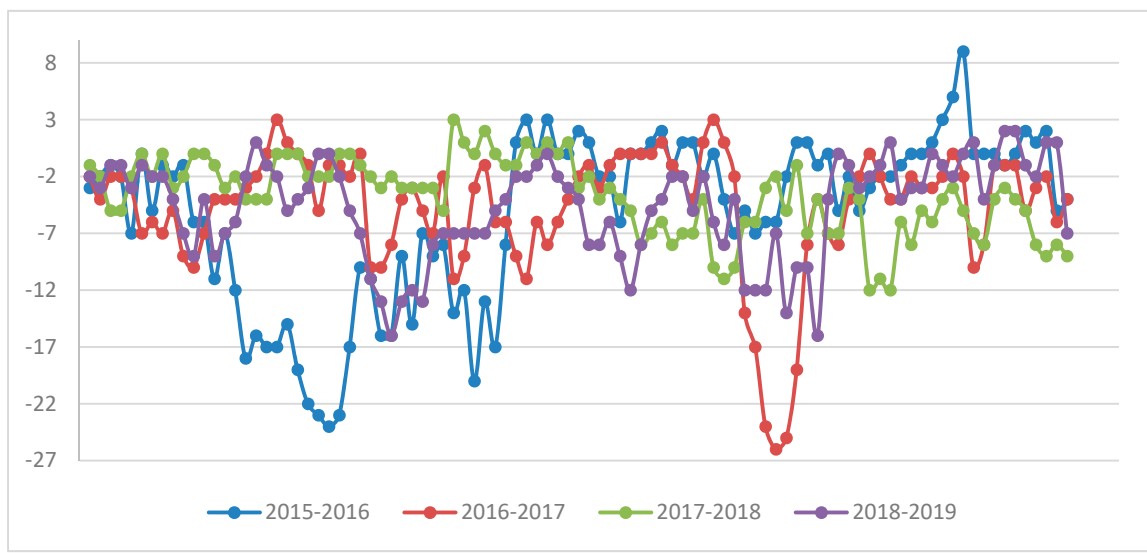

**Figure 3.** Temperature measurement from November to March of each year on the territory of the city of Valday.

According to our data, during the sampling period, the height of the snow column over the indicated four years at all points gradually increased from 25–30 to 40–50 cm, as well as the total precipitation from 90–100 to 140–150 mm. The years 2016–2017 were characterized by the lowest amounts of snow. However, the total volume of meltwater (Table 1) varied within two liters, which may be an indirect indicator of the difference in the chemical composition of the samples. Our assumptions are confirmed by previous studies regarding the trend toward climate warming in the reference (conditional reference) areas of Greenland, Spitzbergen [13–16], which is accompanied by a change in the snow chemical composition. A greater amount of snowfall, together with chaotic periods of freezing/melting, reduced the average concentrations of polluting components due to elements dilution and diffusion with an overall reduced density of the snow column. According to our data, these conclusions are confirmed. In 2016–2017, large contents of the main cationic–anionic composition were revealed as having two- to three-fold greater conductivity. This issue will be considered in more detail below.

3.1.2. Physicochemical Features of Snow and Snow Water

As indicated above, the content of elements in snow depends on the amount of precipitation, temperature fluctuations, and physical properties of snow.

As general parameters, the temperature (T) and pH of snow samples were measured in two aggregate states: snowpack (in situ) and melted snow water. The pH values (Table 1) of the unmelted snow ranged on average from 4.0 to 5.5, depending on the sample density. In general, with more intense snowfall, pH increased, which is associated with a higher concentration of absorbed carbon dioxide from the air and the decrease in the amount of active protons of hydrogen [8,10].

In the prepared melted snow, the pH was slightly higher, 5.5 to 7 or more, and varied depending on the location of the sampling point. The pH values of urban snow samples (points 6 and 7) exceeded 6, associated with the effects of the spread of dust particles of pollutants from the urban environment [17]. The melted snow of the samples from the forest territory was characterized by a slightly lower pH (within 0.5), which coincides with previous reports [17,18] about the effect of organic substances or technogenic sulfates.

**Table 1.** Basic seasonal and physicochemical parameters of snow.

| Parameter | Snow Sampling Point | | | | | |
|---|---|---|---|---|---|---|
| | **1** | **2** | **3** | **4** | **5** | **6** |
| V solution, L | 4.7<br>3.0–5.5 | 4.9<br>4.3–5.3 | 5.1<br>4.7–5.3 | 4.1<br>3.0–5.1 | 5.4<br>4.5–6.5 | 5.2<br>4.6–6.2 |
| T snow, °C | 0.31<br>−0.70–0.67 | 0.18<br>−0.70−−0.61 | 0.25<br>−0.40–0.60 | 0.74<br>0.10–2.10 | 0.87<br>0.20–2.0 | 1.25<br>1.10–4.00 |
| pH solution | 6.13<br>5.76–6.50 | 5.98<br>5.5–5.73 | 6.13<br>5.83–6.65 | 5.93<br>5.70–5.85 | 6.93<br>6.40–7.44 | 6.40<br>6.37–6.82 |
| pH snow | 4.95<br>3.96–6.50 | 5.15<br>4.26–6.50 | 5.05<br>4.00–6.65 | 4.75<br>4.50–5.00 | 6.76<br>5.86–7.44 | 5.80<br>4.16–6.69 |
| Color, ° Cr/Co | 40.3<br>33.4–47.74 | 33.53<br>29.78–39.21 | 25.87<br>17.20–29.78 | 30.80<br>30.00–31.27 | 29.00<br>27.60–30.44 | 24.68<br>13.65–29.12 |
| Turbidity, ° Farm. | 12.00<br>3.00–17.27 | 8.20<br>6.96–9.75 | 4.90<br>3.00–6.40 | 4.60<br>2.60–6.60 | 5.99<br>2.92–8.5 | 4.43<br>1.50–7.73 |
| Cond, µS/cm | 9.9<br>8.1–14 | 8.6<br>7.3–13.1 | 7.6<br>4.9–12 | 8.15<br>8–8.3 | 10.4<br>6.5–15.6 | 9.81<br>4.1–11.3 |
| Si, µg/L | 42.2<br>12–69 | 38.1<br>21–50 | 28.7<br>19–40 | 26.4<br>20–33 | 49.5<br>40–57 | 48.8<br>15–60 |
| Alk, µeq/L | 37<br>3–60 | 18.5<br>3.2–37 | 21<br>3.5–40 | 15<br>4–20 | 31<br>5–64 | 28<br>6–50 |
| Ca, µg/L | 664.5<br>439–666 | 510.5<br>332–591 | 438<br>385–505 | 358<br>333–377 | 575<br>271–950 | 652<br>365–937 |
| Mg, µg/L | 94.7<br>68–107 | 85<br>48–121 | 55<br>30–94 | 62<br>50–75 | 86<br>44–121 | 65<br>32–100 |
| Fe, µg/L | 53<br>16.3–92.5 | 22.2<br>10.4–31.2 | 17.3<br>0–18.0 | 28.1<br>17–39 | 45.5<br>18–84 | 25.5<br>6.7–42.0 |
| Al, µg/L | 28<br>17–33 | 21<br>19–25 | 13.5<br>9–19 | 18.7<br>14–23 | 37.8<br>20–61 | 17.5<br>7.0–22.0 |
| Cu, µg/L | 9.3<br>1.8–20 | 18.4<br>1.5–63 | 5.1<br>2–11 | 4.7<br>3.6–6 | 3.3<br>1.1–6 | 2.65<br>1.2–3.3 |
| Pb, µg/L | 2.6<br>1.2–4.3 | 1.9<br>1.4–2.3 | 2.34<br>1.0–4.0 | 1.6<br>1.4–1.7 | 1.6<br>1.0–2.5 | 2.34<br>1.8–3.4 |
| Ni, µg/L | 1.1<br>0.5–2.15 | 5.8<br>0.8–19 | 1.1<br>0.7–1.8 | 2.4<br>0.6–4.2 | 2.4<br>0.4–5.0 | 1.7<br>1.2–3.0 |
| Cd, µg/L | 0.07<br>0.03–0.1 | 0.06<br>0.06–0.11 | 0.07<br>0.03–0.13 | 0.06<br>0.05–0.06 | 0.06<br>0.06–0.07 | 0.05<br>0.04–0.06 |
| Zn, µg/L | 25.6<br>15–34 | 43.5<br>18–106 | 22<br>19–26 | 25<br>19–31 | 41<br>33–56 | 24<br>13–36 |
| Mn, µg/L | 72.6<br>24–116 | 51.2<br>17.6–94.5 | 8.6<br>4.3–16 | 20<br>7–33 | 6.7<br>4.8–10.8 | 2.7<br>2.1–3.6 |
| Cl, mg/L | 0.55<br>0.15–1.3 | 0.39<br>0.32–0.57 | 0.53<br>0.23–0.80 | 0.45<br>0.27–0.57 | 0.55<br>0.23–0.85 | 0.63<br>0–1.8 |
| SO$_4$, mg/L | 0.6<br>0.64–0.68 | 0.75<br>0.37–1.6 | 0.40<br>0.30–0.57 | 0.60<br>0.41–0.8 | 0.5<br>0.22–0.65 | 0.45<br>0.22–0.89 |
| P, µg P/L | 40<br>5.3–64.0 | 35<br>0.37–1.6 | 12<br>3.0–19 | 15<br>5.0–56 | 21<br>7.0–35 | 32<br>1.8–80 |
| N, µg N/L | 43<br>30–70 | 37<br>30–60 | 42<br>25–60 | 28<br>20–30 | 39<br>20–70 | 44<br>20–70 |
| ANC, µeq/L | −7.12<br>−16–3.0 | −2.1<br>−17–11 | −1.15<br>−9–2.5 | −7.6<br>−9−−5 | −11<br>−4−−20 | −3.7<br>−2–2 |

### 3.1.3. Content of Organic Substances and Related Parameters

According to the data, under-crown snow water of forest samples was the most enriched with organic substances. Due to the contribution of low molecular weight organic substances and the biological component, an increased color of about 40 degrees, as well as turbidity of about 10–12 was revealed for point 1. Points 2–4 in the forest were characterized not only by close pH values, but also by lower contents of organic substances (about 30) and turbidity (under 8). In comparison with other samples, the color of the urban snow samples was below 30 degrees Cr/Co, with lower turbidity (under 6). Si is an indirect indicator of the content of fine particles [18]. Si concentrations in these samples were high in forest sampling points under coniferous trees (about 40 μg/L) and in urban samples (about 50 μg/L). In the first case, we assumed the organic component and lower pH influenced the formation of polysilicic acids; the second case was influenced by the urban environment and dust particles [13].

Despite the approximately same electrical conductivity values (average data), the maximum values for the samples in different locations varied. The highest measured conductivity values were found in forest snow samples near coniferous vegetation (up to 13–14 μS/cm) and in urban snow samples (up to 10–16 μS/cm). The results obtained are comparable with a previous report [19] and are connected with the contribution of the balance of sulfate, chloride anions of anthropogenic and geochemical, and alkaline earth metal ions [20].

### 3.1.4. Biogenic Elements (N and P)

The biogenic element's contents were the highest in the forest samples. Depending on the proximity to coniferous trees or the influence of vegetation (dry vegetation under snow), both parameters were more than three times higher than those of the lake snow sample. For urban samples, the N content is generally higher than the P content, although the biological component may affect the concentration of phosphates. Elevated concentrations of N in the city may be associated with local urban sources [21].

### 3.1.5. Sulfates and Chlorides

The dominant parameters of the anion composition are characterized by both natural and anthropogenic input routes [22,23]. According to our data, the average concentrations of both parameters did not vary much amongst all sampling points. The slight increase in the proportion of chlorides in the urban samples may be associated with the influence of urban infrastructure [8,10]. The maximum sulfate content (about 1–2 μg/L) was observed both in the forest (points 2–4) and urban samples (points 5–6) with minimal vegetation (open space and dry vegetation under snow). In the forest (point 1), the contribution to concentration was potentially due to the washing off from coniferous vegetation; in the urban area, a transboundary sulfate input was possible.

### 3.1.6. Aluminum, Iron, and Manganese

The influence of geochemical factors on the ingress of aluminum and iron manifested in high concentrations in the forest and urban snow samples in areas with dry vegetation under snow [5,6,24]. The pH of under-crown snow precipitation was lower than the pH of urban samples. Aluminum and iron migrate more actively in a more acidic environment; therefore, their contents in under-crown precipitation were higher [10]. In the urban sample, the increased content of iron and aluminum can be explained by the large open space for active atmospheric enrichment.

Changes in manganese concentrations aligned with a decrease in color (organic matter): small values were characteristic for snow in the forest area and the smallest was observed for urban samples, which is consistent with previous reports [5,6,15,16].

### 3.1.7. Heavy Metal Ions

Despite a sufficient variation in the metal content in the samples and the absence of a clear trend, the largest accumulation of copper, nickel, and zinc was found in a forest snow sample (point 2), which

was characterized by sloped terrain. Similar trends were identified by others [17,25]. No distribution features were found for lead and cadmium due to the small variation in concentrations and total concentration values.

### 3.2. Statistical Analysis of the Snow Chemical Composition

To consider the degree of difference in a large number of parameters, a canonical analysis was used, where T mean; solution pH; color; alkalinity; sulfates; Si; ions of aluminum, iron, lead, copper, cadmium, nickel, zinc, and manganese; and total P were chosen as discriminating parameters.

The data obtained are presented in Figure 4. The visualization shows that the snow samples at points 5 and 6 i.e., the snow near the city of Valday are quite distinguished. Sulfate ions contributed the most to the differences between forest and urban snow samples (f-coefficient 11.3), followed by nickel ions (8.3). Between points was influenced by the tree crown and open space (P, 10.1; color, 9.8).

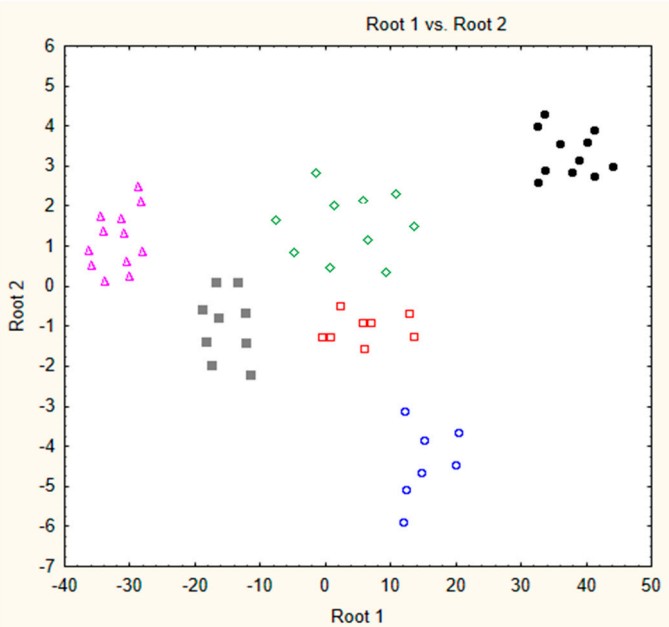

**Figure 4.** Discriminant analysis with the canonical representation of data: point of sampling. Sampling point.

To identify the dominant factors that determined the content of metal ions in the snow, a PCA analysis with RDA graphics was conducted. The biogenic and technogenic parameters of the system, as well as weather factors (pH, color, total sulfur, total phosphorus, electrical conductivity, measured snow temperature, and volume of melted snow) were chosen as determining variables.

According to Figure 5, the following trends were identified: ions of heavy metals such as copper, nickel, lead, as well as hydrogen protons are characterized by a significant affinity for sulfur, which can be explained by both local pollution sources and maybe transboundary migration in different degrees for each point. Possible transboundary metal migrations were considered in various foreign studies [25–28]; they indicated elevated metal contents even in the reference areas of the Arctic and Greenland.

A fairly wide group of cations (manganese, iron, aluminum) and a number of related elements from the group of lanthanides and actinides have an affinity for both organic matter (color) and biogenic component (phosphorus), which is associated with the inclusion of components into biological cycles and their role in redox, complex-forming processes.

A number of elements showed a clear affinity for the representative of biogenic parameters—phosphorus. For example, the presence of elements of the actinide and molybdenum group relative to the P-axis is explained by the geochemical features of the landscape and biological processes.

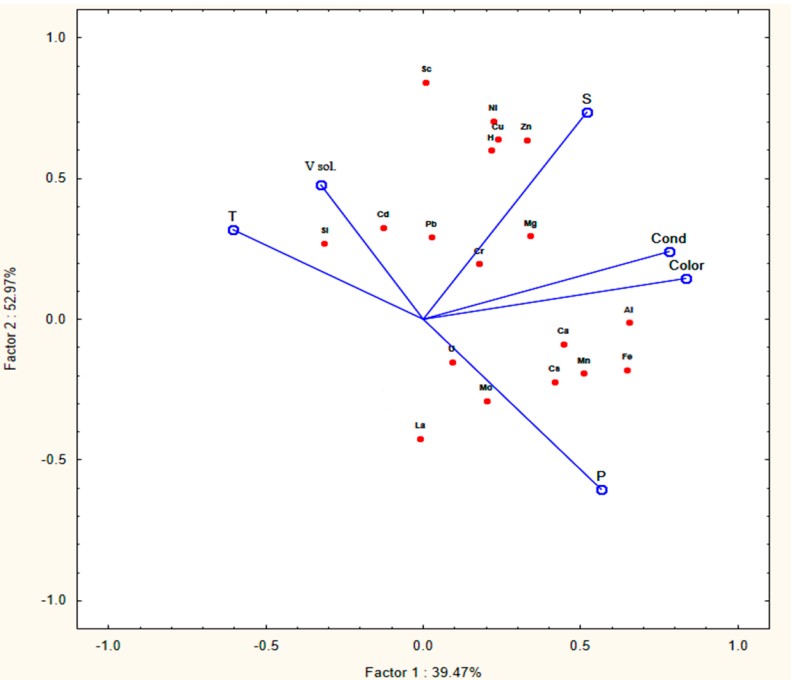

**Figure 5.** Redundancy analysis based on dependent (points) and independent (axis) variables.

A number of elements also showed an affinity for the external parameters of the environment, i.e., the temperature of the snow measured during sampling and the resulting volume of the solution after melting. In the case of lead and cadmium ions, an affinity for the volume of melted snow was observed in addition to an affinity for sulfur-containing components, which may be indirect evidence of the air transfer of these pollutants. AN affinity for temperature and volume was revealed for Si ions, which can be explained by different diffusions of the element from the snow surface into the soil and, therefore, different concentrations in the snow column.

### 3.2.1. Snow Buffer Capacity (ANC)

The balance between cations and anions allowed us to estimate the buffer capacity of snow samples to the effects of transition metal ions capable of proton forming, as well as to the direct effects of acidified atmospheric precipitation [28,29]. ANC is calculated by:

$$ANC = Ca^{2+} + Mg^{2+} + Na^+ + K^+ - SO_4^{2-} - NO_3^- - Cl^- \tag{1}$$

The performed calculations showed (Figure 2) that the acid–based properties of snowpack lack neutralizing compounds, whereas hydrogen ions predominate the melted snow wind transfer of snow, which corresponds to the general situation in Russia [29,30].

According to our data, the mechanisms of equilibrium displacement in the systems are different. Table 1 shows that the minimum values of ANC are rated as follows:

Point 5 (City, −11) > Point 4 (Lake, −7.6) > Point 1 (Forest, −7.12)

The low ANC value in the forest snow sampled near coniferous trees is associated with the influence of low molecular weight organic substances of biological origin. The lake snow samples are close to the previous sample in terms of buffer capacity. The chemical composition of snow is mainly influenced by the sufficiently large open space for atmospheric precipitation and the possibility of snow transfer on the surface of a frozen lake. In this case, both the biological input of protons due to the wind transfer of snow on the surface of the lake ice and the effect of pollutants transfer over long distances, are possible.

The lowest buffer capacity value was found for the urban samples (−11), which clearly indicates the influence of both local technogenic factors and transboundary transfers.

### 3.2.2. Metal Speciation

To assess the speciation of elements in snow samples, both membrane filtration (Figure 6) and chromatographic separation into unstable/stable states were conducted. Intermediate speciation of metals (within 0.45 and 0.2 µm), indicating the colloidal content, showed that mainly organic states of elements [30–32] are the most informative.

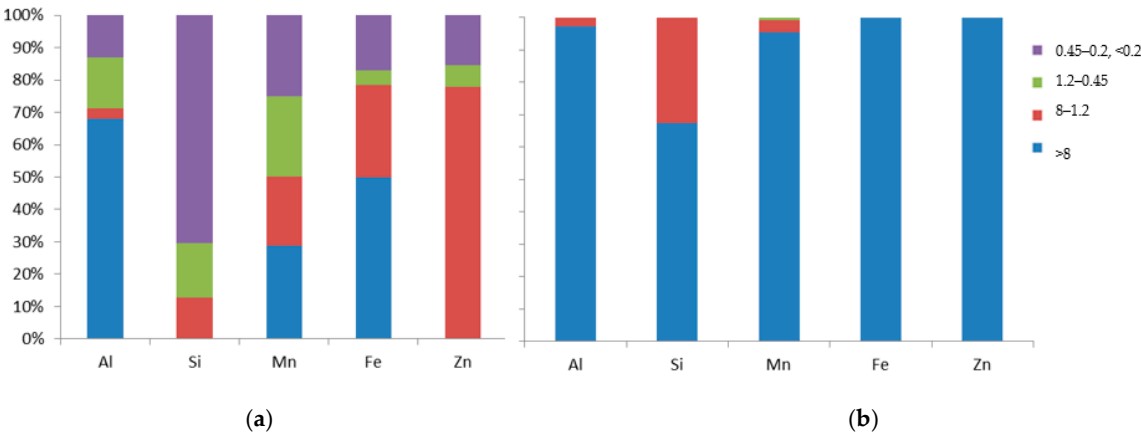

(**a**)　　　　　　　　　　　　　　　　　　　(**b**)

**Figure 6.** The percentage of different metals specification (size—µm). (**a**) Sample 2 (forest); (**b**) Sample 6 (city, near State hydrological institute).

Considering the aluminum and iron compounds, we observed a variety of intermediate colloidal speciation specifically in forest samples and an almost complete shift of these metals in urban samples to the suspended state. The same tendency is common for Si; in snow that is more acidified with organic substances, Si more actively passes into the dissolved polymer state. Heavy metal ions continue the indicated trend toward the most diverse copper speciation—organic complex compounds of various sizes (not more than 1.2 microns).

Despite the low content in the samples, the elements of the lanthanide and actinide series mimicked the identified distribution trends of iron and aluminum.

The ion exchange separation allowed us to determine the conditional stable and unstable speciation of metals. The content of organic matter varied quite strongly in two selected points in the urban area. Stable compounds did not exceed 1–5% of the initial concentration for iron and aluminum ions and less than 1% for copper.

In forest snow, in the presence of organic substances of different molecular weights (an organic matter of mixed forest trees), stable speciation of metals were as follows: Fe > Al > Cu > Ni > Pb.

## 4. Conclusions

The chemical composition of snowfall in the selected area formed under the influence of geochemical, biogenic, and technogenic factors, and was also determined by the sampling point location. From November–March each year in the period 2016–2019, an increase in the average temperature of more than 1 °C occurred, as well as an increase in the atmospheric precipitation volume was found. Despite the short observation period, we observed a trend of climate warming during this period. As the amount of precipitation increases, the content of chemical components in it decreases due to the dilution and diffusion movement of elements.

The influence of coniferous trees and dry vegetation under the snow (points 1 and 2) was associated with the increasing content of organic matter and biogenic elements, and the decreasing pH value. Spreading technogenic pollutants (sulfates and ions of some heavy metals (nickel, copper)) affect such

remote points. On the slopes of an open location (points 3 and 4), subject to the greatest wind transfer of snow, the contents of metals along with Si and alkaline earth elements were the lowest (Si near 26 µg/L, Mg near 60 mg/L). Snowpack near the small city of Valday (points 5 ad 6) reflected increased precipitation of nitrogen and sulfur-containing components from the atmosphere.

Multivariate statistical analysis showed significant differences ($p$ value = 0.000001; F-rem > 8) between the six selected sampling points in terms of sulfate and nickel ions and the biogenic components. In addition, the multivariate axial analysis showed the affinity of metals for various environmental parameters: heavy metals for sulfur; biogenic metals to organic matter and phosphorus. Additionally, a diagnostic sign of 3ew.,43-ntj76yfd7mthe possibile influence of pollution spread was the increased content of metals of technogenic origin, including cadmium, lead, and nickel in the snowpack at an open place.

Using parallel and sequential microfiltration methods, we found that forest snow samples contain metal compounds with different molecular weights due to the different contributions of organic substances. The predominant type of metals in urban snow samples were large particles (mainly more than 8 mkm).

The calculation of the buffer capacity of snow revealed cation deficiency states and negative ANC values, which are common for the European territory of Russia, indicating the distribution of acid-forming agents and their precipitation from the polluted atmosphere in a reference area.

Despite the complexity and ambiguity of the object, the studies showed the influence of geochemical features on the content of nutrients, organic matters in the snow of the forest; the influence of technogenic factors on the content of components (heavy metals, sulfates, turbidity) in the snow of the city (local city pollution), and snow in open areas (airborne migration).

**Author Contributions:** M.D.—expeditionary research, theoretical data analysis, article structure formation; T.M.—theoretical data analysis; D.B.—expeditionary research, geological characteristics of landscapes. All authors have read and agreed to the published version of the manuscript.

**Funding:** Financial support of the Russian Science Foundation grant 18-17-00184.

**Conflicts of Interest:** The authors declare no conflict of interest.

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
