# Peer review of "Snowpack as Indicators of Atmospheric Pollution: The Valday Upland"

_atmosphere, doi:10.3390/atmos11050462_

Round 1

Reviewer 1 Report

The research deals with physico-chemical analysis of snowpack in the Valday Upland (Russian Federation) in 2016-2019. After analyzing a variety of parameters the authors conclude that the study area reflects the regional or even global pollution level.

I would like to point out the following:

  • My overall impression of the text is that it is scientifically sound, but overloaded with technical data. Perhaps some methods that were used according to the literature do not need further explanation; only a reference.
  • The text needs proof-reading as there some errors in presentation (for instance, lines 158, 162, 251, 356, etc.)
  • English language needs revision. Some examples: lines 12-13, 253, etc.
  • Also, all terminology needs to be checked carefully.

Section-by section review:

Introduction

  • The Introduction is well written but I think some statements need citations, specifically, Lines 22-23 and 46-47.
  • Please explain the term ‘technophilic’. You meant ‘technogenic’?
  • Why are certain elements ‘of significant interest for atmospheric pollution assessment’?
  • Please make sure all abbreviations are explained the first time you use them. See Line 40.
  • Why is the Valday Upland an important area for studying environmental pollution?

Materials and Methods

  • Are 6 sampling points enough for this type of research?
  • Please remove Russian captions in figure 1. A map showing the general study area and relative distance between two sites (Gusinoye Lake and Valday city) would help the readers to understand the scale better.
  • I think too much time is spent in this section describing common analysis methods, for example the analysis of hydrochemical parameters. This section would benefit from revision.

Results and discussion

  • A definite merit of this section is that the authors meticulously compare their results with the available literature data.
  • Figure 3 needs some explanations, a legend.

Conclusions are adequate for the study.

Author Response

The authors are grateful  reviewer for the attention to the manuscript and comments!!!!!

The authors tried to take into account all comments as much as possible.

Reviewer 2 Report

The paper highlights chemical composition of snow samples collected in 2016 - 2019 at the Valdai hills. Undoubtedly, the article will be interesting to readers of Atmosphere. But some revisions should be made:

1. Please, provide information from Lines 76 – 96 in Table.

2. Figure 1 needs addition: the small-scale map of sampling area location should be provided.

3. Please, present information from Lines 109 – 131 in Table indicated used equipment and methods with references.

4. It seems to be replication in Lines 110 – 133 and 132 – 137.

5. Figure with the scheme of filtration is very useful to understand all the processes described in Lines 144 – 168. It helps avoiding repeats in Lines 148 – 154 and 162 – 168.

Specific comments are in the manuscript.

Author Response

(The authors gave the same response as above.)

Round 2

Reviewer 2 Report

I have read the revised version of the paper. I think that the manuscript has been improved significantly and now warrants publication in Atmosphere.

Author Response

Dear reviewer. Thank you for your attention to the manuscript and valuable comments. We are completing the improvement of the English language and preparing the article for publication